# miRNAs and Their Gene Targets—A Clue to Differentiate Pregnancies with Small for Gestational Age Newborns, Intrauterine Growth Restriction, and Preeclampsia

**DOI:** 10.3390/diagnostics11040729

**Published:** 2021-04-20

**Authors:** Angelika V. Timofeeva, Ivan S. Fedorov, Alexander G. Brzhozovskiy, Anna E. Bugrova, Vitaliy V. Chagovets, Maria V. Volochaeva, Natalia L. Starodubtseva, Vladimir E. Frankevich, Evgeny N. Nikolaev, Roman G. Shmakov, Gennady T. Sukhikh

**Affiliations:** 1Kulakov National Medical Research Center of Obstetrics, Gynecology, and Perinatology, Ministry of Health of Russia, Ac. Oparina 4, 117997 Moscow, Russia; is_fedorov@oparina4.ru (I.S.F.); a_brzhozovzkiy@oparina4.ru (A.G.B.); or anna.bugrova@gmail.com (A.E.B.); v_chagovets@oparina4.ru (V.V.C.); m_volochaeva@oparina4.ru (M.V.V.); or n.starodub@phystech.edu (N.L.S.); v_frankevich@oparina4.ru (V.E.F.); r_shmakov@oparina4.ru (R.G.S.); g_sukhikh@oparina4.ru (G.T.S.); 2Laboratory of Mass Spectrometry, CDISE, Skolkovo Institute of Science and Technology, 121205 Moscow, Russia; e.nikolaev@skoltech.ru; 3Emanuel Institute of Biochemical Physics, Russian Academy of Sciences, 119334 Moscow, Russia; 4Department of Chemical Physics, Moscow Institute of Physics and Technology, 141700 Dolgoprudny, Moscow Region, Russia; 5Department of Obstetrics, Gynecology, Neonatology and Reproduction, First Moscow State Medical University Named after I.M. Sechenov, 119991 Moscow, Russia

**Keywords:** intrauterine growth restriction, preeclampsia, small for gestational age, placental bed, placenta, miRNA deep sequencing, mass spectrometry, protein, mRNA, reverse transcription, polymerase chain reaction in real time

## Abstract

Despite the differences in the clinical manifestations of major obstetric syndromes, such as preeclampsia (PE) and intrauterine growth restriction (IUGR), their pathogenesis is based on the dysregulation of proliferation, differentiation, and invasion of cytotrophoblast cells that occur in the developing placenta, decidual endometrium, and myometrial parts of the spiral arteries. To understand the similarities and differences in the molecular mechanisms of PE and IUGR, samples of the placental bed and placental tissue were analyzed using protein mass spectrometry and the deep sequencing of small RNAs, followed by validation of the data obtained by quantitative RT-PCR in real time. A comparison of the transcriptome and proteomic profiles in the samples made it possible to conclude that the main changes in the molecular profile in IUGR occur in the placental bed, in contrast to PE, in which the majority of molecular changes occurs in the placenta. In placental bed samples, significant changes in the ratio of miRNA and its potential target gene expression levels were revealed, which were unique for IUGR (miR-30c-5p/VIM, miR-28-3p/VIM, miR-1-3p/ANXA2, miR-30c-5p/FBN1; miR-15b-5p/MYL6), unique for PE (miR-185-3p/FLNA), common for IUGR and PE (miR-30c-5p/YWHAZ and miR-654-3p/FGA), but all associated with abnormality in the hemostatic and vascular systems as well as with an inflammatory process at the fetal‒maternal interface.

## 1. Introduction

Abnormal placentation due to multiple causes is considered a key pathogenetic mechanism of various pregnancy complications, particularly preeclampsia (PE) and intrauterine growth restriction (IUGR) [1]. As a consequence, structural and functional defects of the placenta predispose to a variety of chronic diseases after delivery and increase the morbidity and mortality of mother and fetus [2,3,4]. PE occurs in 2–8% of pregnancies worldwide and is characterized by hypertension, proteinuria, and/or edema, with additional signs of multiple organ failure depending on its form: mild, moderate, or severe [5,6]. The late-onset type of PE, with clinical manifestation after 34 gestation weeks (GW), comprises more than 80% of all PE cases and is associated with normal or only slightly altered behavior of the uterine spiral arteries and no signs of fetal growth restriction. The early-onset type of PE (clinical manifestation before 34 GW) comprises the most severe cases of PE (5–20%) and is associated with incomplete trophoblast invasion of maternal spiral arteries, resulting in changes in the blood flow in the uterine arteries, placental vessels, and umbilical arteries, with signs of fetal growth restriction.

IUGR is a condition that affects 5–10% of pregnancies, characterized by slow intrauterine growth of the fetus with an expected fetal weight below the 10th percentile, assessed for gestational age and gender, associated with abnormal Doppler ultrasound findings in the vessels of fetus and umbilical cord [7,8]. Early IUGR (diagnosed at or below 32 GW) differs from late-onset IUGR (diagnosed after 32 GW) in terms of its clinical manifestations, association with hypertension, patterns of deterioration, and severity of placental dysfunction [9].

A detailed and systemic analysis of the normal placenta development with an indication of the possible pathogenetic mechanisms of PE and IUGR is presented in the review article by Aplin JD [10]. Taking a look at early human development, the progression from zygote through morula to blastocyst culminates in the differentiation of cells into the trophectoderm, which will form the placenta, and the inner cell mass, which will form the fetus. Trophectoderms of human blastocysts are highly invasive, and to establish a flourishing intrauterine pregnancy, the embryo trophoblast cells must anchor to and invade the decidualized endometrium. Endometrial invasion has two components: (i) interstitial invasion of the extravillous cytotrophoblasts from anchoring villi into the decidua and stroma, and (ii) the endovascular route, in which the vascular remodeling is affected by the migration and invasion of endovascular trophoblasts into the spiral arteries, ensuring an increase in their diameter and a decrease in their resistance [11]. It has been reported that endovascular invasion of trophoblasts occurs in two waves: invasion into the decidual segments of spiral arteries at 8–10 weeks of gestation followed by invasion into myometrial segments at 16–18 weeks of gestation [12]. In parallel, the generation of the outer multinuclear syncytiotrophoblast layer occurs due to continuous proliferation and cell fusion of villous cytotrophoblasts, making possible nutrient transport and gas exchange at the interface between mother and fetus.

Any deviations from the normal physiological process, starting with the formation of the trophectoderm during blastulation, followed by the differentiation towards villous and extravillous trophoblast, lead to the development of IUGR or PE. Berthold Huppertz hypothesized about the pathophysiological mechanisms of PE and IUGR [13], assuming that failure in the differentiation of the extravillous trophoblast pathway results in pure IUGR with typical characteristics such as failed invasion and an abnormal uterine artery Doppler. PE may result from failure of the villous pathway, with typical characteristics such as the release of trophoblastic fragments, which may cause systemic alterations of the maternal endothelium and inflammatory response. However, if the villous as well as the extravillous pathways of trophoblast differentiation are affected, in particular during trophoblast development, a combination of PE and IUGR arises, as is often the case with early-onset severe PE.

Epigenetics likely plays a prominent role in embryogenesis, the main regulators of which are small noncoding RNAs (sncRNAs) [14,15,16,17]. These molecules, in particular, miRNAs, function as a part of the RNA-induced silencing complex (RISC) affecting the integrity of RNA target by the endonuclease activity of Argonaute proteins and/or blocking protein synthesis by the interaction with translation initiation factors [18]. In our previous studies [19,20], we found significantly changed levels of sncRNAs in the spent culture media from embryos with different developmental outcomes and implantation potential. We have shown that the sncRNA expression profile and their potential gene targets at the morula stage reflect the efficiency of maternal‒zygotic transition and blastulation while forming embryoblasts and trophoblasts. Despite the abundance of scientific reports on the role of miRNAs in the development of IUGR and PE, summarized and discussed in Hu’s review article [21], up to date there is no information on the key experimentally proven pathogenetic participants of the “miRNA-target gene” network, which differ in IUGR and PE. In this connection, and in order to test Huppertz’s hypothesis, we compared the transcriptome and proteomic profiles in the placenta tissue from the fetal to maternal surfaces as well as in the placental bed tissue from the decidual surface to the myometrial base, containing extravillous trophoblast cells and myometrial parts of the spiral arteries, from pregnant women with PE, IUGR, and small for gestational age (SGA) newborns.

## 2. Results

### 2.1. Clinical Characteristics of Patients with PE, IUGR, and SGA and Healthy Pregnant Women

Clinical laboratory evaluation of pregnant women with PE, IUGR, and SGA and healthy pregnant women is summarized in Table 1. Since the patients in the groups with IUGR and PE were delivered before or after 34 weeks of gestation, the comparison of the clinical parameters of the patients was performed both within these groups (IUGR < 34 vs. IUGR > 34; PE < 34 vs. PE > 34) and each of the groups was compared to a gestational age-matched control (IUGR > 34 vs. N > 34; PE > 34 vs. N > 34; SGA > 34 vs. N > 34). There were no differences between patient groups in terms of maternal age, body mass index (BMI), pulsatility index (PI) of middle cerebral artery (MCA), or PI of left uterine artery. There were significant differences between patient groups with complicated pregnancies and the control group in terms of blood pressure, APGAR 1 min and 5 min, indicator of fetal condition according to cardiotocography (CTG) data, PI of right uterine artery, birth weight, placental weight, PI of umbilical artery, percentile of estimated fetal weight, gestational age at the time of delivery, and cerebroplacental ratio. Estimated fetal weight below the 10th percentile, significant increase of PI of umbilical artery and PI of right uterine artery, significant decrease of placenta mass, and intraventricular hemorrhage in 12.5–20% of cases were observed only in pregnant women with IUGR and/or SGA. A significant increase in blood pressure and proteinuria in 57.14–83.3% of cases were observed only in pregnant women with PE. Patients with a significant decrease in the cerebral placental ratio were observed in the groups with SGA, IUGR, and PE. The increase in the indicator of fetal condition according to CTG was characteristic of groups of patients with IUGR and PE.

### 2.2. miRNA Expression Signature in Placenta and Placenta Bed Tissue Samples

To identify differentially expressed miRNAs in patient groups (PE < 34, *n* = 3 out of 12; IUGR < 34; *n* = 3 out of 10) compared to a gestational age-matched control (N < 34, *n* = 3), deep sequencing of miRNAs from placenta and placental bed samples was carried out, followed by data processing and analysis in the DESeq2 program [22]. The results of this comparison are presented in Appendix A (Sheet 1 for placenta samples, PE/N; Sheet 2 for placenta samples, IUGR/N; Sheet 3 for placental bed samples, PE/N; Sheet 4 for placenta bed samples, IUGR/N). Venn diagrams were plotted for the comparative analysis of the miRNA differential expression patterns as follows: placenta vs. placenta bed samples from patients with PE (Figure 1a); placenta vs. placenta bed samples from patients with IUGR (Figure 1b); placental bed samples from patients with PE vs. IUGR (Figure 1c); placenta samples from patients with PE vs. IUGR (Figure 1d).

In PE, the greatest changes in the level of miRNA expression occur in the placenta, which mainly consists of the chorionic plate and chorionic villi, compared with the placental bed (Figure 1a), which is composed of a decidual layer, extravillous cytotrophoblasts, and myometrial parts of the spiral arteries. These results emphasize the important role of the placental villi, covered by the syncytiotrophoblast layer in the pathogenesis of PE. In contrast with IUGR, the greatest changes in the level of miRNA expression occur in the placental bed compared with the placenta (Figure 1b), which emphasizes the important role of the maternal‒fetal interface in the pathogenesis of IUGR.

At the same time, the intersection of lists of differentially expressed miRNAs in samples of placental bed from patients with PE and IUGR (Figure 1c) was not observed. On the contrary, when comparing the expression profiles of miRNA in placenta samples from patients with PE and IUGR versus control placenta samples (Figure 1d), we found a total of 10 miRNAs (upregulated hsa-miR-16-2-3p, hsa-miR-146b-5p, hsa -miR-29b-1-5p, hsa-let-7g-5p, hsa-miR-27a-5p; downregulated hsa-miR-4497, hsa-miR-7704, hsa-miR-3940-3p, hsa-miR-204-5p, hsa-miR-4286), which apparently reflects the generality of changes in the regulation of the activity of signaling pathways under the control of this group of miRNAs in the placental tissue from women with PE and IUGR.

### 2.3. Protein Expression Profiles in Placenta and Placenta Bed Tissue Samples 

The same tissue samples of the placenta and placental bed used for deep sequencing of miRNAs were analyzed by label-free semiquantitative HPLC-coupled mass spectrometry. The obtained protein profiles in the placenta and placental bed from patients with IUGR < 34 and PE < 34 were compared with those in control samples (N < 34) using the MaxQuant (Max-Planck-Institute of Biochemistry, Munich, Germany) bioinformatics package (version 1.6.7.0). The results of the analysis are presented in Appendix A (Sheet 1 for placenta samples, PE/N; Sheet 2 for placenta samples, IUGR/N; Sheet 3 for placental bed samples, PE/N; Sheet 4 for placental bed samples, IUGR/N). Venn diagrams were plotted for the comparative analysis of the protein differential expression patterns as follows: placenta vs. placenta bed samples from patients with PE (Figure 2a); placenta vs. placenta bed samples from patients with IUGR (Figure 2b); placental bed samples from patients with PE vs. IUGR (Figure 2c); placenta samples from patients with PE vs. IUGR (Figure 2d)).

A comparison of the resulting lists of differentially expressed proteins by Venn diagrams revealed a larger group of proteins with altered levels of expression in placenta samples, rather than in placental bed samples, from women with PE (Figure 2a). In contrast, in IUGR, the greatest changes in the level of protein expression occur in the placental bed compared with the placenta (Figure 2b). The obtained data of mass spectrometric analysis of proteins repeated the trend of changes in the placenta and placental bed tissues from women with IUGR and PE. In addition, 29 proteins specific to IUGR and seven proteins specific tot PE were identified in the placental bed samples (Figure 2c). A smaller group of differentially expressed proteins that distinguish IUGR from PE was found in the placenta (Figure 2d). However, significant changes in protein expression in both the placenta (Figure 2d) and the placental bed (Figure 2c) were common to PE and IUGR.

### 2.4. Bioinformatic Search for miRNA–Target Gene Pairs that Differentiate IUGR from PE

To date, the molecular biological mechanisms of the pathogenesis of IUGR and PE have not been fully studied. Therefore, to search for their differences, the objects of our further detailed study were samples of the placental bed, in which the expression profiles of miRNA and proteins significantly distinguished IUGR from PE (Figure 1c and Figure 2c). For this purpose, first of all, the microT-CDS algorithm was used in the DIANA-miRPath v3.0 program (http://snf-515788.vm.okeanos.grnet.gr/index.php?r=mirpath, (accessed on 15 September 2020)) to search potential target genes for miRNAs differentially expressed in placental bed samples from patients with IUGR relative to control samples (Appendix A, Sheet 4). The obtained list of target genes was compared with the list of differentially expressed proteins according to our mass spectrometry data from the same placental bed samples (Appendix A, Sheet 4). Thus, the list of potential target genes with proven differential expression in the placental bed samples from patients with IUGR was compared with the list of miRNAs (Appendix A, Sheet 4) with the opposite direction of changes in the level of expression in the same placental bed samples. The miRNA–target gene pairs formed in such a way are presented in Table 2.

### 2.5. Assessment of the Expression Level of miRNAs and Their Target Genes in the Placental Bed Samples by Quantitative RT-PCR

The data of semiquantitative protein mass spectrometry and miRNA deep sequencing were validated by RT-PCR in the placental bed samples from the entire cohort of patients (*n* = 57), which included pregnant women with (1) IUGR delivered before 34 GW (IUGR < 34); (2) IUGR delivered after 34 GW (IUGR > 34); (3) PE delivered before 34 GW (PE < 34); (4) PE delivered after 34 GW (PE > 34); (5) small for gestational age newborns, delivered after 34 GW (SGA > 34); and (6) physiological course of pregnancy, delivered after 34 GW (N > 34). Clinical and instrumental data of the patients of each group are presented in Table 1.

We found significant differences in the expression level of certain miRNAs and protein-coding genes within the IUGR (IUGR < 34 vs. IUGR > 34) and PE groups (PE < 34 vs. PE > 34), depending on the time of delivery (Figure 3a,b, respectively; Table 3). The dependence of the miRNA expression level on the gestational age and the need to compare the analyzed samples from pregnant women with age-matched samples was demonstrated in our previous study [23]. Samples from women with preterm delivery earlier than 34 GW, often due to pelvic inflammatory disease, are not optimal controls for age-matched samples from pregnant women with IUGR or PE. These findings are based on the research of Paquette et al., who identified transcriptomic changes associated with spontaneous preterm labor at 24–34 GW in maternal whole blood and peripheral monocytes compared to healthy pregnant women matched for gestational age who subsequently delivered at term [24]. It is known that monocytes and different populations of macrophages (proinflammatory M1 and anti-inflammatory M2 macrophages) are involved in placental invasion, angiogenesis, and tissue remodeling. In addition, complicated pregnancies—in particular, PE—are characterized by altered functional activity of the monocyte–macrophage system [25]. Therefore, in order to understand the molecular mechanisms of the pathogenesis of pregnancy complications, it is more informative and correct to compare placental bed samples from healthy women, delivered after 34 GW (N > 34), with age-matched samples from women with IUGR > 34 (Figure 3c, Table 3), PE > 34 (Figure 3d, Table 3), or SGA > 34 (Figure 3e, Table 3). In addition, we aimed to compare IUGR, PE, and SGA in the same GW range. Since all women from the SGA group were delivered after 34 GW, for further comparison with this group, the groups with IUGR > 34, PE > 34, and control group N > 34 were selected, but not other groups (IUGR < 34, PE < 34).

A significant decrease in the expression level of eight out of 11 analyzed miRNAs and an increase in the level of six out of 11 analyzed mRNAs in the IUGR > 34 relative to N > 34 in the placental bed samples were revealed (Figure 3c, Table 3). Among them were those with the opposite direction of change in the level of expression that formed a miRNA‒gene target pair, namely: hsa-miR-1-3p/ANXA2; hsa-miR-30c-5p/VIM; hsa-miR-28-3p/VIM; hsa-miR-654-3p/FGA; hsa-miR-30c-5p/YWHAZ; hsa-miR-1-3p/YWHAZ; hsa-miR-30c-5p/FBN1; and hsa-miR-15b-5p/MYL6, which also formed in IUGR < 34 (Table 2). In addition, a significant decrease in the expression level of hsa-miR-199a-3p and hsa-miR-199b-3p, which are potential regulators of the FN1 gene, and a significant decrease in the level of expression of hsa-miR-140-3p, potentially regulating DSTN, were found in the group IUGR> 34 (Figure 3c, Table 3). At the same time, no significant changes in the level of FBN1 mRNA and DSTN mRNA were found. It can be assumed that changes in the expression level of these genes in the IUGR > 34 group occur only at the translation level, since significant changes in the proteins of the FBN1 and DSTN genes were found in the IUGR < 34 group by mass spectrometry (Table 2). Alternatively, it may be the case that there are no changes in the expression level of these genes at all in the IUGR > 34 group.

A significant decrease in the level of expression of hsa-miR-30c-5p, hsa-miR-654-3p, and hsa-miR-185-3p, and an increase in the mRNA level of the *MYH11*, *FGA*, *FLNA*, and *YWHAZ* genes were revealed in the placental bed samples from the PE > 34 group relative to N > 34 (Figure 3d, Table 3). Some of them with the opposite direction of change in the level of expression formed the following miRNA‒target gene pairs: hsa-miR-654-3p/FGA; hsa-miR-30c-5p/YWHAZ; and hsa-miR-185-3p/FLNA. When comparing the PE > 34 group with the IUGR > 34 group, the common pairs were hsa-miR-654-3p/FGA and hsa-miR-30c-5p/YWHAZ, while the hsa-miR-185-3p/FLNA pair was unique for PE > 34, and hsa-miR-1-3p/ANXA2, hsa-miR-30c-5p/VIM, hsa-miR-28-3p/VIM, hsa-miR-1-3p/YWHAZ, hsa-miR-30c-5p/FBN1, and hsa-miR-15b-5p/MYL6 were unique pairs for IUGR > 34, which probably indicates both commonality and differences in the pathogenesis of the two obstetric syndromes.

A significant decrease in the expression level of hsa-miR-654-3p, hsa-miR-15b-5p, hsa-miR-185-3p, and hsa-miR-140-3p and an increase in the level of FGA mRNA and MYL6 mRNA were detected in placenta bed samples from SGA > 34 relative to N > 34 (Figure 3e, Table 3), forming the following miRNA‒target gene pairs with the opposite direction of changes in the expression of its components: hsa-miR-654-3p and FGA; and hsa-miR-15b-5p and MYL6. These pairs were found to be common for the SGA > 34 and IUGR > 34 groups, which probably indicates the commonality of changes in the maternal‒fetal interface that affect the anthropometric parameters of the fetus.

We presume that it is not informative enough to separately analyze the molecules that form particular signaling pathway, involved in the pathogenesis of pregnancy complications. It is the comparison of the ratios of the expression levels of the regulatory molecule and its target gene in normal and pathological tissues that can be the key to understanding the dysregulation of this signaling pathway. Therefore, it seemed interesting to calculate the ratios of the expression level of miRNAs and their potential target genes in each of the studied samples to search for differences between the groups with IUGR > 34, PE > 34, and SGA > 34 with reference to N > 34. This interest was also supported by the correlation analysis data on the expression level of miRNA and their target genes (Figure 4). It is worth noting the significant inverse correlation of the expression level of molecules that form miRNA‒target gene pairs, namely: hsa-miR-1-3p and ANXA2, hsa-miR-1-3p and YWHAZ, hsa-miR-30c-5p and VIM, hsa-miR-30c-5p and YWHAZ, and hsa-miR-28-3p and VIM.

In light of the above, the difference between the ΔCt miRNA and ΔCt mRNA values in each sample was found, the 2^-ΔΔCt^ values were calculated, followed by its logarithm base two. The medians of the obtained values in each group are presented in Table 4, which is the base 2 logarithm of the median of the ratio of the miRNA and mRNA expression levels.

In IUGR > 34, a significant decrease in the ratio of the expression levels of miRNA and its target gene relative to N > 34 was found in nine pairs, PE > 34 differed from N > 34 in a reduced ratio of miRNA and its target gene in three pairs, and the group SGA > 34 differed from N > 34 by a reduced ratio of miRNA and its target gene in only one pair (Table 4). At the same time, despite the fact that the miR-654-3p/FGA pair significantly distinguished all the analyzed groups from the control group, the decrease in the ratio of the expression levels of miR-654-3p and FGA mRNA was more pronounced in the SGA and PE groups than in the IUGR group. The miR-30c-5p/YWHAZ pair significantly distinguished the PE and IUGR groups from the N group, but the decrease in the ratio of miR-30c-5p and YWHAZ mRNA expression levels was more pronounced in the IUGR group than in the PE group relative to the control group. The decrease in the ratio of expression levels of the miRNA and mRNA unique for the IUGR group relative to the control group was demonstrated for miR-28-3p and VIM mRNA, miR-15b-5p and MYL6 mRNA (transcription variant 2), miR-15b-5p and MYL6 mRNA (transcription variant 1), miR-1-3p and ANXA2 mRNA, miR-1-3p and YWHAZ mRNA, miR-30c-5p and FBN1 mRNA, and miR-30c-5p and VIM mRNA. Unique for the PE group was a decrease in the ratio of miR-185-3p and FLNA mRNA expression levels relative to the control group.

### 2.6. Partial Least Squares Discriminant Analysis (PLS-DA) 

The PLS-DA model based on RT-PCR data was developed to study differences between healthy pregnant women and women with IUGR, PE, or SGA. The logarithm to base two of fold change expression level of miRNA relative to its target gene, calculated as a (−ΔΔCt) value in each of the samples, was used for the model. The list of the analyzed miRNA‒target gene pairs is presented in Table 2. The contribution of miRNA‒gene target pairs to the distribution of the data points on the score plots of the developed PLS-DA models was estimated by the Variable Importance in Projection (VIP) score for IUGR group and control group (Figure 5a), SGA group and control group (Figure 5b), PE group and control group (Figure 5c). The score plots of the developed PLS-DA models are presented as an insert in the upper right corner of each of the VIP score plots, Figure 5a–c). The miRNA‒gene target pairs with VIP > 1 and the highest impact for the differentiation of IUGR from N were as follows: miR-15b-5p/MYL6 tr.v.2 (VIP = 1.28), miR-15b-5p/MYL6 tr.v.1 (VIP = 1.24), miR-654-3p/FGA (VIP = 1.22), miR-30c-5p/YWHAZ (VIP = 1.16), miR-30c-5p/VIM (VIP = 1.14), miR-30c-5p/FBN1 (VIP = 1.11), and miR-28-3p/VIM (VIP = 1.03). In the insert of Figure 5a representing the score plot of the developed PLS-DA model, two clusters of data points can be distinguished. The first one (highlighted in red) has an abscissa of less than 0.0 and represents the samples of placental bed from women with normal pregnancy except one sample. The second cluster (highlighted in blue) has an abscissa of more than 0.0 and represents the samples of placental bed from women with IUGR except one sample.

The highest impact for the differentiation of SGA from normal (Figure 5b) had the following pairs: miR-654-3p/FGA (VIP = 1.98), miR-15b-5p/MYL6 tr.v.1 (VIP = 1.07), and miR-140-3p/DSTN (VIP = 1.06). The division into clusters on the score plot of the developed PLS-DA model (insert in Figure 5b) was not as obvious as in the case of the IUGR and N comparison, which may indicate small differences between the SGA group and the control group, with the exception of the anthropometric characteristics of the fetus.

The miRNA‒gene target pairs with a VIP value > 1 separating the samples of the PE and N groups from each other (Figure 5c) were as follows: miR-654-3p/FGA (VIP = 2.00), miR-185-3p/FLNA (VIP = 1.17), miR-128-3p/FN1 (VIP = 1.08), and miR-30c-5p/YWHAS (VIP = 1.05). Samples from the PE and N groups were not clearly distinguished from each other, as follows from the score plot of the developed PLS-DA model (the insert of Figure 5c). This is explained by the fact that the analyzed pairs were selected on the basis of their differential expression in the placental bed in the IUGR relative to the control (Table 2), but not in any other complications of pregnancy.

### 2.7. Functional Annotation of miRNA Target Genes

To assess the functional significance of the studied miRNA target genes, a Metascape enrichment analysis of proteins from Table 2 was carried out using the Gene Ontology and Reactome databases. The functional relationships between the proteins as a subset of enriched terms are presented in Table 5.

It follows from Table 5 that ANXA2, VIM, FGA, MYL6, YWHAZ, and FBN1, which significantly distinguish the IUGR group from the control group (Figure 3c and Figure 5a, Table 3 and Table 4), are involved in the processes associated with signaling by interleukins, activation of platelets and their degranulation, hemostasis, blood vessel development and morphogenesis, smooth muscle contraction, angiogenesis, and extracellular matrix organization. According to the published data, the starting point for obstetrical syndromes such as IUGR and PE is abnormality in the hemostatic and vascular systems as well as an inflammatory process in the maternal and fetal segments of the placenta [26].

For the given gene list, protein‒protein interaction enrichment analysis has been carried out in Metascape with the following databases: BioGrid6, InWeb_IM7, and OmniPath8. The resultant network contains the subset of proteins that form physical interactions with at least one other member in the list. The Molecular Complex Detection (MCODE) algorithm 9 has been applied to identify densely connected network components, as shown in Figure 6.

Figure 6 clearly shows that there are common components in the pathogenesis of IUGR, PE, and SGA, as well as molecular mechanisms specific to each obstetric syndrome. So, unique to IUGR are the changes in the expression of hsa-miR-30c-5p, hsa-miR-28-3p, and their gene-target VIM; hsa-miR-1-3p and ANXA2; hsa-miR-30c-5p and FBN1; hsa-miR-15b-5p and MYL6. Changes in the expression level of the hsa-miR-185-3p/FLNA pair were found to be unique for PE, while quantitative changes in the hsa-miR-30c-5p/YWHAZ and hsa-miR-654-3p/FGA ratios were common to IUGR and PE. The hsa-miR-654-3p/FGA ratio also changed in the SGA group. Since all the proteins shown in Figure 5 physically interact with each other directly or indirectly, the degree of severity of placental bed changes in IUGR, PE, and SGA will be different depending on the degree of regulatory influence of the corresponding miRNA on the expression level of a particular protein, which determines the clinical signs of these syndromes.

## 3. Discussion

In the present study, we applied different approaches such as small RNA deep sequencing, RT-PCR, and mass spectrometry to compare the miRNA and protein coding gene expression profiles, and found that the greatest changes at the transcriptomic and proteomic levels in IUGR occur mainly in the placental bed, in contrast to PE, where most of the changes occur in the placental tissue. These data are consistent with the hypothesis of Huppertz, assuming that failure in the differentiation of the extravillous trophoblast pathway results in pure IUGR in contrast to PE, which may result from failure of the villous pathway with its typical characteristics such as systemic alterations of the maternal endothelium and inflammatory response [13]. Impaired trophoblast differentiation by the extravillous pathway results in inadequate remodeling of the maternal spiral arteries, limited artery obstruction by endovascular trophoblasts up to 10 weeks of gestation, and, as a consequence, an increase in the exposure of the placental villi to injury from reactive oxygen and nitrative species [27]. Normal early placental development occurs in an environment with a low oxygen tension to limit oxidative stress, but removal of the blockage of the spiral arteries occurs only from the 10th week of pregnancy, with the placental villi directly bathed with oxygen- and nutrient-rich maternal blood until delivery. Maladaptation of the spiral arteries induces mechanical damage of the trophoblastic villi due to the increased pressure of blood entering the intervillous space [28]. Villi injuries under mechanical and oxidative stress affect signaling pathways mediated by nuclear factor κB and p38 MAPK, resulting in inflammation, dysregulation of the trophoblast differentiation, and diminished placental growth, which altogether manifest as placental insufficiency [29,30].

Placental malperfusion in the maternal segment is known to lead to abnormal uterine blood flow, while malperfusion in the fetal segment of the placenta increases velocimetry indices in the umbilical artery, which have been linked to increased cardiovascular and central nervous system problems in newborns [31,32,33,34]. Our data demonstrate the signs of placental insufficiency in the IUGR > 34 group of pregnant women by (i) increased pulsatility index of right uterine artery, (ii) increased pulsatility index of umbilical artery, (iii) decreased cerebroplacental ratio as a response of the fetus to hypoxia in the form of redistribution of the blood toward the cerebral circulation achieved by cerebral vasodilatation, (iv) decreased placenta mass, (v) deterioration of the fetus according to CTG, and (vi) fetal weight below 10th percentile. We confirmed ultrasound-detectable hemodynamic changes by significant alterations in expression level of miRNAs and their gene targets associated with extracellular matrix organization, blood vessel development and morphogenesis, smooth muscle contraction, angiogenesis, endothelial damage, and thrombi formation according to the Gene Ontology and Reactome databases. These miRNA‒gene target pairs are as follows: hsa-miR-30c-5p, hsa-miR-28-3p and their gene target VIM; hsa-miR-1-3p and ANXA2; hsa-miR-30c-5p and FBN1; hsa-miR-15b-5p and MYL6; and hsa-miR-30c-5p/YWHAZ and hsa-miR-654-3p/FGA. The data obtained are consistent with the general histological characteristics of the placenta with IUGR, which have significantly increased maternal and fetal vascular lesions compared to placentas from normal pregnancies: maternal vascular lesions occur in about 50% of placentas from women with IUGR at term versus only 20% in normal pregnancies, while fetal vascular lesions occur in 11% of IUGR pregnancies versus 4% in normal pregnancies [26]. Lesions of the maternal vascular compartment are related to maternal underperfusion (acute atherosis, increased syncytial knots, increased intervillous fibrin deposition, and villous infarcts). Placental fetal vascular lesions are the result of stasis, hypercoagulability, and vascular damage within the fetal circulation of the placenta. It is underlined that vascular lesions in placenta of IUGR pregnancies are often associated with chronic inflammatory processes, which is supported by the Metascape enrichment analysis data on the involvement of ANXA2/VIM/YWHAZ in signaling by interleukines, suggesting that more than one mechanism is involved in the development of IUGR.

The major intermediate filament protein in endothelial cells that is regulated by hypoxia, altering contractility and the adhesiveness of the endothelium, is vimentin [35]. It is a marker of epithelial to mesenchymal transition, classically involved in maintaining cell structure, polarity, cell-to-cell adhesion, and cellular motility [36]. It is considered a potential master regulator of choriodecidual genes and associated with inflammation at term labor [37]. According to the small RNA sequencing and protein mass spectrometry data presented here, decreased expression of vimentin and increased expression of its potential regulators, miR-30c-5p and miR-28-3p, were found in placental bed samples in the group IUGR < 34 relative to the age-matched control group. On the contrary, an increased level of vimentin mRNA and decreased expression of miR-30c-5p and miR-28-3p were found in placental bed samples in the IUGR > 34 group relative to the control group N > 34, according to RT-PCR data. Differences in the ratio of the expression level of miR-30c-5p/VIM and miR-28-3p/VIM in the IUGR < 34 and IUGR > 34 groups may be due to the different severity of changes in the placental bed in these obstetric syndromes and/or the dependence of the expression level of these miRNAs and vimentin on the gestational age. However, it is important to note the opposite changes in expression levels of miR-30c-5p, miR-28-3p/VIM, and their target gene, vimentin, in each of the IUGR groups and the absence of any expression changes of these molecules in the SGA > 34 and PE > 34 groups. According to the literature data, studies on the expression of vimentin in the IUGR placental bed are absent, but the expression of vimentin in the preeclamptic placentas remains controversial: an increased expression of vimentin in the intravillous area was observed in the placentas of preeclamptic patients with HELLP [38], in contrast to decreased mRNA and protein expression levels of vimentin in the placental tissues of patients with preeclampsia, as shown by Du et al. [39]. 

In light of the general theories on IUGR and PE pathogenesis described above, fibrinogen is of particular interest as it plays a pivotal role in several physiological settings, such as coagulation and inflammation [40]. It is essential for sustaining the development of fetal‒maternal circulation [41], for supporting early-term trophoblast proliferation and spreading [42], and for the development and maintenance of the placenta [43]. In our study, the level of mRNA of the fibrinogen alpha gene (FGA) was found to be increased in the placental bed from patients with SGA > 34, PE > 34, and IUGR > 34, but, at the same time, the decrease in the ratio of the expression levels of miR-654-3p and FGA mRNA was more pronounced in the SGA and PE groups than in the IUGR group relative to the control group. These data may indicate differences in the epigenetic control, in particular under the influence of miR-654-3p, of the FGA expression level in the placental bed in IUGR, PE, and SGA.

The fact that the placentas in obstetrical syndromes often display various infarctions and fibrin deposits reflects defects of fibrinolytic system that predispose to increased thrombosis [44]. The increased thrombin generation observed among these patients may derive not only from increased activation of the hemostatic system, but also from insufficient anticoagulation, as reflected by the lower tissue factor pathway inhibitor (TFPI) concentrations. It was shown that patients with PE, eclampsia, and IUGR have a lower placental concentration of total TFPI than women with normal pregnancies [45,46]. The overall balance between the coagulation factors—in particular, tissue factors (TF) and their inhibitors (one of which is TFPI)—increases thrombin generation. Indeed, the TFPI/TF ratio of these patients was lower than that of normal pregnant women, mainly due to the decreased TFPI concentrations. In contrast, the maternal plasma TFPI concentration does not change in mothers with SGA fetuses [47].

In turn, fibrinolysis is a well-organized cascade process through interaction of different components of fibrinolytic system among which the principal molecules are plasminogen activators such as tissue-type plasminogen activator (tPA), urokinase-type plasminogen activator and their inhibitors [48]. Homeostatic control of plasmin activity is exerted through receptor-mediated binding of different components of the fibrinolytic system [49]. For example, annexin A2 (ANXA2) is the endothelial cell receptor for tPA and plasminogen, and this assembly promotes and localizes cell surface plasmin generation [50]. Within this context, the expression of ANXA2 plays a key role in maintaining the fibrinolytic balance on the blood vessel endothelium [51]. It has been shown that complete deficiency of ANXA2 in mice is associated with microvascular fibrin accumulation and impaired clearance of injury-induced arterial thrombi [52]. It was found that placental ANXA2 mRNA expression was significantly elevated in a group of PE cases with acute worsening of symptoms necessitating immediate pregnancy termination compared with the normal group, but did not change in a group of PE cases without emergency termination of pregnancy [53]. These data are in good agreement with our data on the absence of significant changes in ANXA2 in the placental bed samples in the group of PE > 34 relative to age-matched controls. At the same time, a significant increase in the expression level of ANXA2, along with a significant decrease in the expression level of miR-1-3p, which is its potential regulator, was found for the first time in the placental bed samples in the group of IUGR > 34 in comparison to the control group N > 34. No changes in the expression levels of miR-1-3p and ANXA2 were found in the group SGA > 34. Previously, ANXA2 has also been identified in the syncytiotrophoblast [54] and was shown to be upregulated by hypoxia [55], leading to the suggestion that it is part of a defense system against hypoxia [56]. Thus, a significant decrease in the miR-1-3p/ANXA2 expression level ratio due to a decrease in miR-1-3p expression and an increase in ANXA2 expression in the placental bed in the group of women with IUGR > 34 can be a compensatory mechanism to prevent increased thrombogenesis, as well as a consequence of hypoxia caused by insufficient placental perfusion. In contrast, a significant increase in the miR-1-3p/ANXA2 expression level ratio due to an increase in miR-1-3p expression and a decrease in annexin A2 expression in the placental bed in the group of women with IUGR < 32 can reflect microvascular fibrin accumulation and, as a result, increased thrombogenesis, necessitating immediate pregnancy termination.

A higher apoptosis rate in the placental tissues, as well as in the cultured cytotrophoblasts of patients with PE and IUGR, compared to the controls is well known [57]. Clusters of nuclei with condensed chromatin typical of apoptosis, so-called syncytial knots, are characteristic of the histopathology in the syncytiotrophoblast of the villous tree in placenta from these patients in response to placenta malperfusion, hypoxia and villous injury [58]. One of the regulators of apoptotic processes in the cell are 14-3-3 proteins, which are highly conserved cytoplasmic molecules involved in the protection of the cell against apoptosis by sequestering different pro-apoptotic proteins [59]. Among the various isoforms of protein 14-3-3, a key molecule in the suppression of apoptosis in the placenta is 14-3-3 zeta (YWHAZ), which interacts with and isolates Bax. It was found that the expression of YWHAZ in preeclamptic placentas was significantly higher compared to normal placentas, but in spite of it, the interaction of YWHAZ with Bax was significantly lower in preeclamptic placentas due to 14-3-3 zeta phosphorylation by PKC delta [60]. These data are consistent with the results obtained in the present study, where we demonstrated a significant increase in the YWHAZ mRNA level and a decrease in the miR-30c-5p expression level in the PE > 34 group relative to the N > 34 group. It should be noted that the miR-30c-5p/YWHAZ pair also significantly distinguished IUGR > 34 from N > 34, but the decrease in the ratio of miR-30c-5p and YWHAZ mRNA expression levels was more pronounced in the IUGR group than in the PE group relative to the control group. This may be due to the impaired additional epigenetic control of the expression level of the YWHAZ by miR-1-3p, which was found to be downregulated in the IUGR > 34.

In addition, it is possible that obstetric syndromes associated with impaired placentation are predetermined by the quality of the embryo itself during early embryogenesis. In our previous studies [19,20], we have shown that sncRNA expression profile and their potential gene targets at the morula stage reflect the efficiency of maternal‒zygotic transition and blastulation while forming embryoblasts and trophoblasts. These results are consistent with previous reports on the identification of genes associated with blastocyst formation [61], among which was myosin light chain 6 (MYL6) gene. Two transcripts of the MYL6 gene, as a result of alternative splicing, showed a difference in expression in degenerate embryos compared with blastocysts [62]. MYL6 is an essential light chain of myosin II, which is an actin-binding protein expressed in smooth muscle and nonmuscle tissues. In the latter, it plays a central role in the intracellular transport of organelles, signal transduction, cell shape, cell adhesion, migration, and division [63]—key processes associated with implantation and placentation. In addition, it has been proven that the excessive airway narrowing seen in asthmatic patients is associated with an increased level of MYL6 due to the increase in the contractility of airway smooth muscle [64]. In this study, significant opposite changes in the expression level of miR-15b-5p and MYL6 were found in the SGA and IUGR groups relative to the control group, in contrast to the PE group, in which no such changes were found. Moreover, the IUGR group significantly differed from the SGA and PE groups in terms of the ratio of the miR-15b-5p and MYL6 expression levels, which emphasizes the differences in the pathogenesis of IUGR, PE, and SGA. In the case of PE, significant differences in the expression level of another actin-binding protein, filamin A (FLNA), and an opposite change in its regulatory miRNA, miR-185-3p, were found. FLNA crosslinks actin filaments into orthogonal networks in the cortical cytoplasm and participates in the anchoring of membrane proteins for the actin cytoskeleton. In the case of a change in the level of FLNA expression, the cell cytoskeleton is remodeled, which leads to a change in the shape of the cell and its mobility. Filamins also interact with a variety of cellular proteins, in particular, transcription factors, transmembrane receptors, and signaling molecules, thereby providing control over the ability of the cell to respond to various signals and to change cell motility [65,66,67]. Downregulation of FLN-b was found in the PE placenta in trophoblasts and capillary endothelial cells, and its level in late-onset PE was lower than in early-onset PE [68]. In our study, in contrast to PE, we did not find any significant differences in miR-185-3p and FLNA mRNA expression in the IUGR and SGA groups, which reflects differences in the pathogenesis of PE, IUGR, and SGA.

One of the important proteins during decidualization and placentation is fibrillin. In a recent study it has been found to be synthesized by the extravillous trophoblast (EVT) cells and to contribute to increased tissue stiffness by acting as a fiber-reinforced composite [69]. It is known that the endometrium undergoes extensive remodeling during decidualization and invasion by EVT cells. Thus, impaired synthesis of fibrillin-1 (FBN1) by EVT cells can lead to impaired placentation due to altered decidual tissue stiffness. In the present study, in samples of the placental bed, we found significant differences in FBN1 expression between the IUGR < 34 and IUGR > 34 groups, as well as significant differences in the ratio of the expression levels of FBN1 and its regulatory miR-30c-5p between the IUGR > 34 and N > 34 groups. 

While applying the molecular complex detection algorithm, a network of proteins that form physical interactions with each other was found to be implicated in the pathogenesis of IUGR, PE, and/or SGA in the present study (Figure 6). In the center of this network was fibronectin 1, a widely distributed extracellular multidomain glycoprotein considered by some investigators as a “master organizer” in the process of matrix assembly forming a bridge between cell surface receptors, in particular, integrins, and extracellular matrix molecules such as type I and III collagen, thrombospondin-1, heparin, and fibrillin-1 (reviewed in [70]). The methods of protein mass spectrometry and small RNA deep sequencing applied here revealed significant opposite changes in the expression level of fibronectin 1 and its regulatory miR-1-3p, miR-199a-3p, and miR-199b-3p in IUGR < 34. However, no changes in the expression of fibronectin 1 mRNA level were found in IUGR > 34, in contrast to miR-1-3p, miR-199a-3p, and miR-199b-3p, whose levels were significantly altered. Probably, under the influence of these miRNAs, there is a change in the translation of fibronectin 1, but not in the stability of its mRNA. Further research is required to clarify this. However, according to the previous reports, specific changes in fibronectin isoform expressions are dependent on the stage of villous maturation and extravillous trophoblast cells invasiveness [71,72], and placental fibronectin appeared to be unaffected in women with late-onset IUGR and SGA fetuses in contrast to early-onset pre-eclampsia with statistically increased level [73]. As for other molecules that are interconnected components of this network (Figure 5), we have shown for the first time that, depending on the expression level of protein-coding genes and their regulatory miRNAs, one or another obstetric syndrome occurs, in particular, IUGR, PE, or SGA.

The main changes in the transcriptome and proteome profiles in IUGR occurred in the placental bed, in contrast to PE, in which the majority of molecular changes occurred in the placenta.

In placental bed samples, significant changes in the ratio of miRNA and its potential target genes expression levels were revealed and were unique for IUGR (miR-30c-5p/VIM, miR-28-3p/VIM, miR-1-3p/ANXA2, miR-30c-5p/FBN1; miR-15b-5p/MYL6), unique for PE (miR-185-3p/FLNA), and common for IUGR and PE (miR-30c-5p/YWHAZ and miR-654-3p/FGA).

Despite the small number of samples in each of the compared groups (on average, 10 samples), which is a limitation of this study, the strength of the present study is the identification of key pairs “miRNA-target gene”, namely miR-30c-5p/VIM, miR-28-3p/VIM, miR-1-3p/ANXA2, miR-30c-5p/FBN1; miR-15b-5p/MYL6, miR-185-3p/FLNA, miR-30c-5p/YWHAZ and miR-654-3p/FGA, able to differentiate PE, IUGR and SGA by expression level in the placental bed. This approach can be used to verify the diagnosis made by clinical and laboratory research methods on the day of delivery to select the optimal case management for the mother and newborn in order to prevent possible postpartum complications. Owing to the results presented herein, it has become possible to clearly differentiate SGA and IUGR, but meticulous examination and careful monitoring will be required for a child in the postnatal period for the early diagnostics of cardiovascular, pulmonary, neurological, and reproductive postnatal diseases associated with IUGR [2,74,75].

In order to solve long-term problems, obtained data provide the fundamental basis for targeting the key pathophysiological mechanisms underlying IUGR and PE to prevent their development at the stage of pregnancy planning and during pregnancy considering the participation of the revealed “miRNA/gene-target” pairs in the regulation of the system hemostasis, immune system, and angiogenesis.

## 4. Materials and Methods 

### 4.1. Patient Cohort

In total, 57 pregnant women aged between 27 and 37 years with Caesarean section indications were enrolled in the study and comprised the following six groups (Table 1): (1) women with early-onset IUGR delivered at 30–32 GW (*n* = 10); (2) women with late-onset IUGR delivered at 35–38.5 GW (*n* = 13); (3) women with early-onset PE delivered at 29–32 GW (*n* = 12); (4) women with late-onset PE manifesting the disease after 34 GW and delivered at 34–36 GW; (5) women with small for gestational age (SGA) neonates who delivered at 36–38 GW (*n* = 8); (6) women with physiological pregnancy delivered at 35–37 GW due to other obstetric indications that cannot be associated with the etiopathogenesis of tested disorders (*n* = 7); (7) women with an indication for an emergency Caesarean section (*n* = 3, 24–34 GW) without clinical manifestations of IUGR and PE.

PE was diagnosed as described in [76]. IUGR was established according to the ISUOG Practice Guidelines [77]. Blood flow in the uterine, middle cerebral, and umbilical arteries of the fetus was determined using Doppler ultrasonography (Voluson E10, GE Healthcare Technologies, Milwaukee, WI, USA). To evaluate the velocity curves of blood flow, the pulsatility index (PI) was used. To assess the state of the fetus, cardiotocography (CTG) was performed using the automated antenatal monitor AAM-04 (UNICOS, Moscow, Russia) in pregnant women with a gestational age over 33 weeks.

SGA was diagnosed by the presence of the abdominal circumference and estimated fetal weight below 10th percentile for the corresponding gestational age [77].

Exclusion criteria in the study were multifetal pregnancies, severe somatic pathology in pregnant women, fetal aneuploidy, and vaginal delivery.

All pregnant women enrolled in the study were prescribed Elevit Pronatal (Bayer) during the first trimester of pregnancy and Femibion Natalcare II (Merck) during the second and third trimesters of pregnancy. Three out of eight pregnant women with SGA, all pregnant women with early-onset IUGR, and two out of thirteen pregnant women with late-onset IUGR were prescribed 0.4 mL Enoxaparin sodium from 36.6 ± 0.94 GW, 28.2 ± 3.8 GW, and 36 ± 0.25 GW, respectively. All pregnant women with early-onset and late-onset PE were prescribed Methyldopa (up to 1 gramm per day), Nifedipine (up to 20 mg per day), 0.4 mL Enoxaparin sodium from 29.2 ± 2.8 GW and 35 ± 1 GW, respectively.

The study was performed according to the verdict of the Local Ethics Committee of Federal State Budget Institution “Research Centre for Obstetrics, Gynecology and Perinatology” after informed consent was signed by the patients.

### 4.2. RNA Isolation from the Placenta and Placenta Bed Samples

Placenta and placental bed tissue samples were taken for research no later than 10 min after delivery. The placenta tissue was sampled as described in [23] according to the recommendations of Burton et al. [78]. Placental bed wedge-shaped samples of 0.5 cm in length were obtained by scalpel/scissors biopsy at Caesarean section from decidual surface to the myometrial base, containing extravillous cytotrophoblasts and myometrial parts of the spiral arteries. 

Sampled placental tissue free of fetal membranes and placental bed were washed in 0.9% NaCl and immediately frozen in liquid nitrogen for subsequent storage at 80 °C. Total RNA was extracted from 20 to 40 mg of tissue followed by quantitative and qualitative assessment as described in [23]. Total RNA samples with a RIN of at least 8 were used for further studies.

### 4.3. miRNA Deep Sequencing

cDNA libraries were synthesized using 500 ng total RNA from the placenta and placental bed tissues as described in [23]. The sequence reads were mapped to the GRCh38.p15 human genome and miRBase v21 with the bowtie aligner [79] followed by counting the number of aligned reads with the featureCount tool from the Subread package [80] and with the fracOverlap 0.9 option, so the whole read was forced to have a 90% intersection with sncRNA features. Differential expression analysis of the sncRNA count data was performed with the DESeq2 package [22].

### 4.4. Reverse Transcription and Quantitative Real-Time PCR

To quantify the miRNA, 500 ng of total RNA from the placental bed tissue were converted into cDNA in a reaction mixture (20 µL) containing 1× Hispec buffer, 1× Nucleics mix, and miScript RT according to the miScript^®^ II RT Kit protocol (Qiagen, Hilden, Germany); then, the sample volume was adjusted with deionized water to 200 µL. The synthesized cDNA (2 µL) was used as a template for real-time PCR using a forward primer specific for the studied miRNA (Table 6) and the miScript SYBR Green PCR Kit (Qiagen, cat. no 218075). The following PCR conditions were used: (1) 15 min at 95 °C and (2) 40 cycles at 94 °C for 15 s, an optimized annealing temperature (48.9–59 °C) for 30 s and 70 °C at 30 s in a StepOnePlus^TM^ thermocycler (Applied Biosystems, USA). The relative expression of miRNA was determined by the ∆∆Ct method using SNORD68 (Hs_SNORD68_11 miScript Primer Assay, Qiagen) as the reference RNA.

To quantitate mRNA, 500 ng of total RNA were mixed with 250 pmol of a random primer (N6), pre-incubated in a volume of 10 μL at 70 °C for 5 min, followed by immediate cooling on ice, and converted into cDNA in a reaction mixture with a 25 µL final volume containing 1× MMLV buffer (Promega), 1× dNTP (Evrogen), and 200 U MMLV reverse transcriptase (Promega). The reactions were performed at 37 °C for 60 min and stopped by incubation at 95 °C for 5 min with subsequent addition of 3 volumes (75 μL) of the stop solution (1 mM EDTA, 10 mM Tris-HCl, pH 8.0). The cDNA probes were stored frozen at −20 °C. PCR amplification was carried out with 2 μL cDNA in a 20-μL reaction mixture containing 1× qPCRmix-HS (SYBR+HighROX, Evrogen) using forward and reverse primers specific for the studied mRNA (Table 6). The following PCR conditions were used: (1) 15 min at 95 °C and (2) 40 cycles at 94 °C for 30 s, an optimized annealing temperature (48.9–61.6 °C) for 30 s and 70 °C at 40 s in a CFX96TM Real Time System (BioRad, Hercules, CA, USA). The relative expression of mRNA was determined by the ∆∆Ct method using GAPDH and ACTB as the reference RNAs.

### 4.5. Proteomic Tissue Analysis (HPLC-MS/MS)

The homogenized placental tissue was dissolved in RIPA buffer with a mixture protease inhibitor (Roche), centrifuged at 10,000× *g*, with supernatant sampling. A lysis buffer with urea was added to the pellet and centrifuged at 10,000× *g*. The supernatants were pooled, and the protein fraction was precipitated with acetone, washed with chilled ethanol, and resuspended in 8M urea with 0.1% Rapigest Surfactant (Waters Corporation, Milford, CT, USA). The total protein concentration was measured by the bicinchoninic acid method (BCA, Thermo Fisher, Waltham, MA, USA). The disulfide bonds in proteins were reduced by 25 mM DTT and alkylated with 50 mM iodoacetamide. The enzymatic hydrolysis included two steps: 4 h of LysC (1/100) and 16 h of trypsin (1/50). The sample was desalted using ZipTip microcolumns, dried on a centrifugal vacuum evaporator, and dissolved in 0.1% formic acid for further HPLC-MS/MS analysis as previously described [81,82]. The peptide mixture (2 µL) was analyzed in triplicate on a nano-HPLC system Dionex Ultimate3000 system (Thermo Fisher Scientific, Waltham, MA, USA) coupled to a timsTOF Pro (Bruker Daltonics, Billerica, MA, USA) mass spectrometer. Peptides were separated with optimized 120-min HPLC gradient (from 3% to 90% of phase B) at 400 nL/min flow rate. LC-grade water with 0.1% formic acid was used as phase A and LC-grade acetonitrile with 0.1% formic acid was used as phase B. For HPLC separation a packed emitter column was used (C18, 25 cm × 75 µm × 1.6 µm) (Ion Optics, Parkville, Australia). MS analysis was carried out by TIMS TOF Pro mass-spectrometer in the mass range from 100 to 1700 m/z via parallel accumulation serial fragmentation (PASEF) acquisition method. 

Protein identification and semiquantitative analysis were performfed by MaxQuant (Max-Planck-Institute of Biochemistry, Germany) package (version 1.6.7.0) using the forward and reverse versions of the SwissProt database. The MS/MS search included eight main peaks within 100 Da, fixed modification—carbamidomethylation of cysteines; variable modifications—N-terminal acetylation and methionine oxidation; at least seven amino acids per peptide; and a false discovery rate (FDR) for proteins and peptides of less than 0.01. 

### 4.6. Statistical Analysis of the Obtained Data

Statistical differences between groups were examined using nonparametric Welch’s *t*-tests with Benjamini‒Hochberg correction for multiple comparison (*p* < 0.01).

For miRNA deep sequencing and RT-PCR data statistical processing, we used scripts written in the R language [80], and RStudio [83]. The Shapiro‒Wilk test was used to assess the correspondence of the analyzed parameters to the normal distribution law. In the case of the unnormal data distribution, the Mann‒Whitney test for paired comparison was used, and data were presented as the median (Me) and quartiles Q1 and Q3 in the format Me (Q1; Q3). Chi-square test was performed to identify the relationship between categorical variables. Spearman’s nonparametric correlation test was applied for analysis of quantitative and qualitative characteristics. Fisher transformation was used to determine the 95% confidence interval for the correlation coefficient. The value of the threshold significance level *p* was set to 0.05. If the *p* value was less than 0.001, then *p* was indicated in the format *p* < 0.001.

## 5. Conclusions

The main changes in the transcriptome and proteome profiles in IUGR occurred in the placental bed, in contrast to PE, in which the majority of molecular changes occurred in the placenta.

In placental bed samples, significant changes in the ratio of miRNA and its potential target genes expression levels were revealed and were unique for IUGR (miR-30c-5p/VIM, miR-28-3p/VIM, miR-1-3p/ANXA2, miR-30c-5p/FBN1; miR-15b-5p/MYL6), unique for PE (miR-185-3p/FLNA), and common for IUGR and PE (miR-30c-5p/YWHAZ and miR-654-3p/FGA).

The clinical signs of PE and IUGR are apparently determined by the spectrum of miRNA molecules and their target genes and the degree of changes in their expression level associated with abnormality in the hemostatic and vascular systems as well as with an inflammatory process at the fetal‒maternal interface, ultimately leading to diminished placental growth and hypoxic injury, which altogether manifest as placental insufficiency.

## Figures and Tables

**Figure 1 diagnostics-11-00729-f001:**
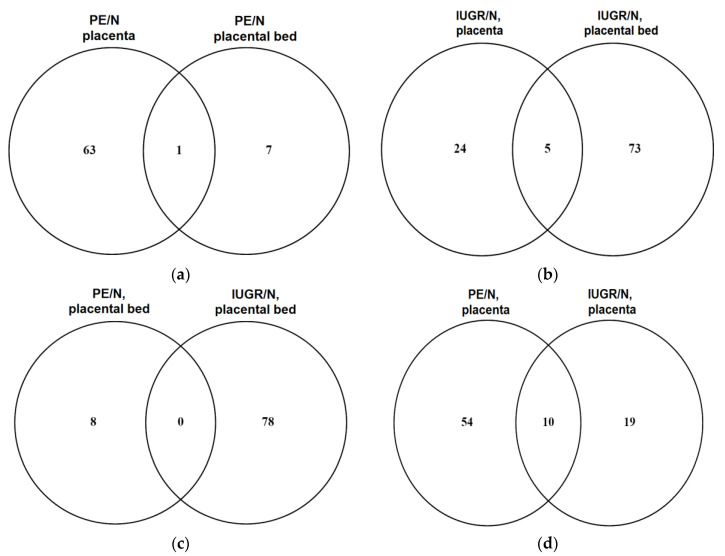
Venn diagrams. A comparison of the lists of differentially expressed miRNAs in (**a**) placenta vs. placenta bed samples from patients with PE < 34; (**b**) placenta vs. placenta bed samples from patients with IUGR < 34; (**c**) placental bed samples from patients with PE < 34 vs. IUGR < 34; (**d**) placenta samples from patients with PE < 34 vs. IUGR < 34.

**Figure 2 diagnostics-11-00729-f002:**
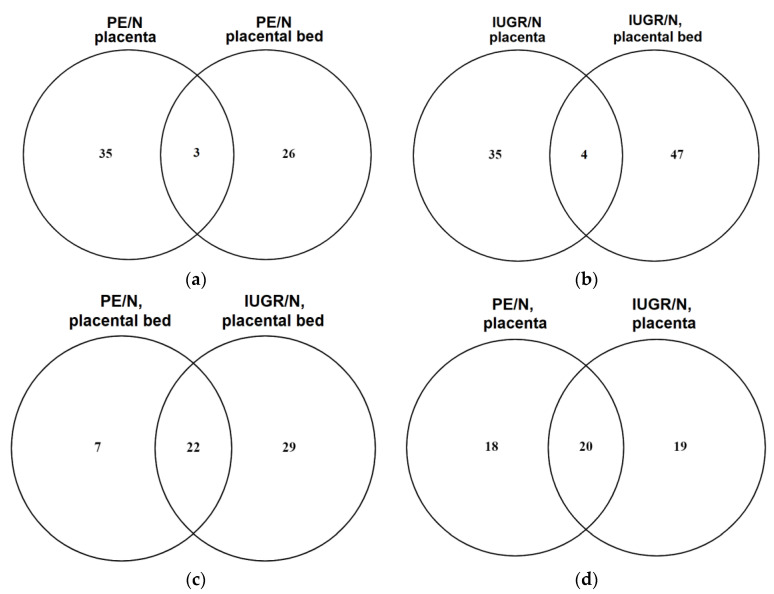
Venn diagrams. A comparison of the lists of differentially expressed proteins in (**a**) placenta vs. placental bed samples from patients with PE < 34; (**b**) placenta vs. placental bed samples from patients with IUGR < 34; (**c**) placental bed samples from patients with PE < 34 vs. IUGR < 34; (**d**) placenta samples from patients with PE < 34 vs. IUGR < 34.

**Figure 3 diagnostics-11-00729-f003:**
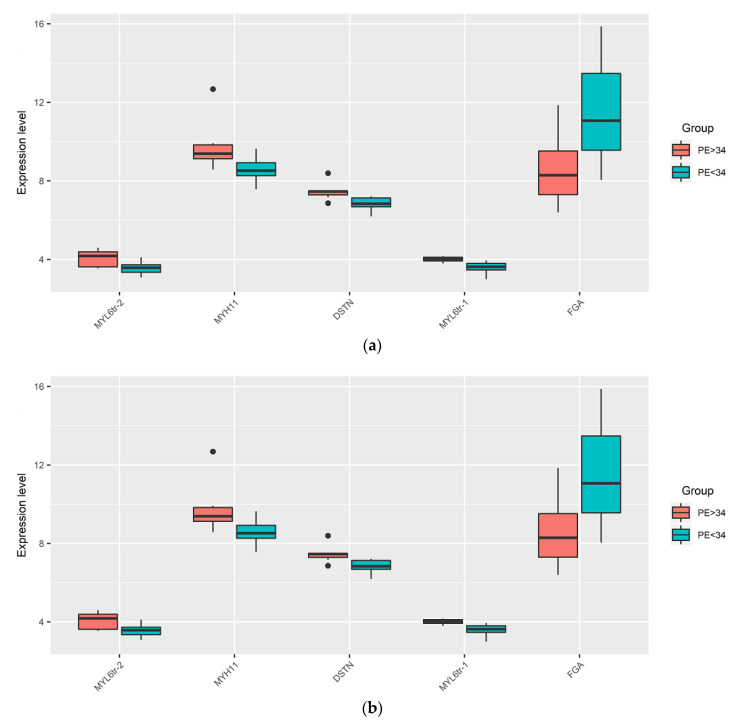
Box-plots of the expression level of miRNAs and their target genes in the placental bed samples by quantitative RT-PCR. Comparison of ∆Ct values of miRNA and mRNA in groups with IUGR < 34 vs. IUGR > 34 (**a**); PE < 34 vs. PE > 34 (**b**); IUGR > 34 vs. N > 34 (**c**); PE > 34 vs. N > 34 (**d**); SGA > 34 vs. N > 34 (**e**).

**Figure 4 diagnostics-11-00729-f004:**
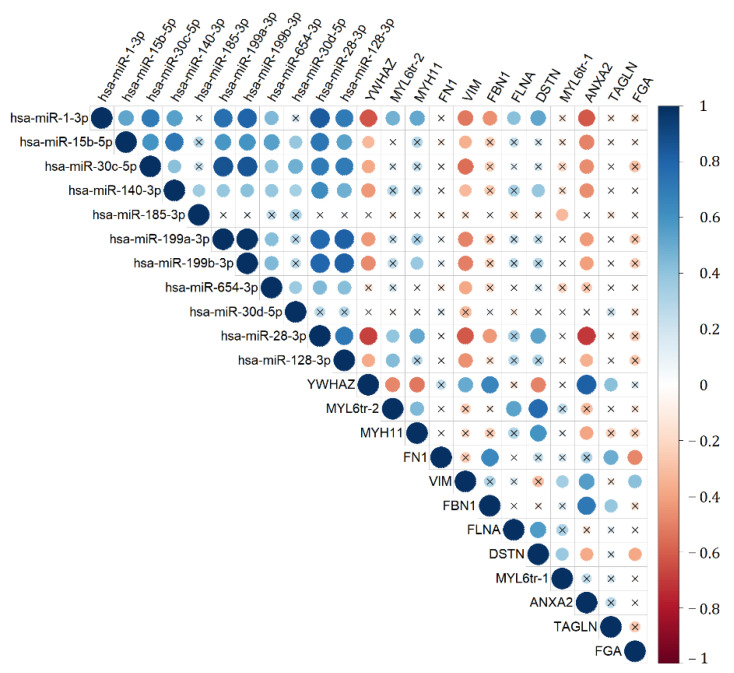
Correlation matrix based on the analysis of the miRNAs and gene target expression levels in the IUGR > 34, SGA > 34, PE > 34, and N > 34 groups obtained while using nonparametric Spearman rank correlation method. Positive and negative correlations are indicated in blue and in red, respectively. In the case of significant (*p* < 0.05) values, correlations are indicated by a dot; in the case of nonsignificant values, correlations are indicated by a cross.

**Figure 5 diagnostics-11-00729-f005:**
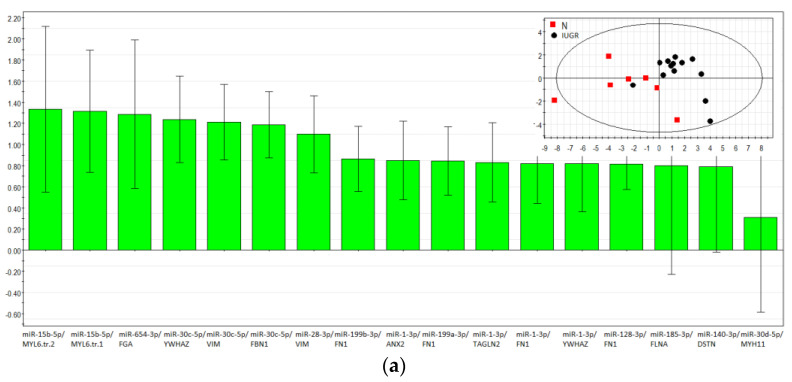
Partial least squares discriminant analysis (PLS-DA). Variable importance in projection (VIP) score plots using (−ΔΔCt) RT-PCR data on the fold change expression level of miRNA relative to its target gene in a sample from the IUGR > 34 and N > 34 groups (**a**), the SGA > 34 and N > 34 groups (**b**), and the PE > 34 and N > 34 groups (**c**). Score plots with the imposition of information of the miRNA‒gene target fold change in the expression level on the sample group type are presented as inserts in the upper right corner of (**a**–**c**).

**Figure 6 diagnostics-11-00729-f006:**
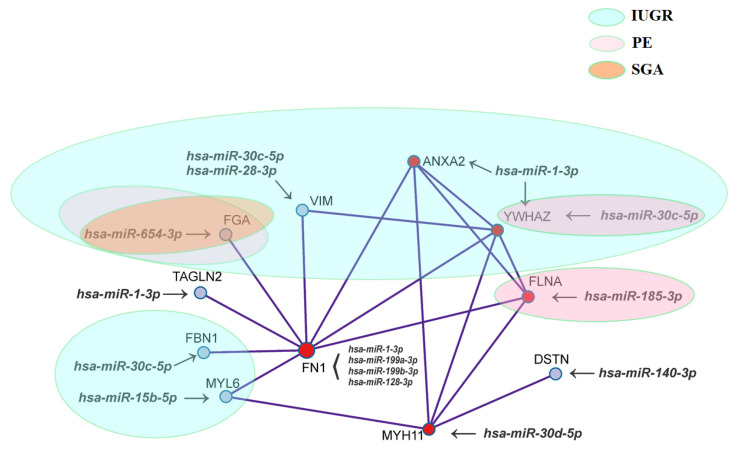
Network of proteins that form physical interactions with each other while applying the Molecular Complex Detection (MCODE) algorithm. Ovals of different colors contain miRNA‒gene target pairs, differentiating the IUGR, PE, and SGA groups from the control group.

**Table 1 diagnostics-11-00729-t001:** Parameters of patients’ clinical characteristics, ultrasonography, and Doppler sonography.

Clinical Characteristics	IUGR < 34 *n* = 10	IUGR > 34 *n* = 13	PE < 34 *n* = 12	PE > 34 *n* = 7	SGA > 34 *n* = 8	N > 34 *n* = 7	*p*-Value: IUGR < 34 vs. IUGR > 34 PE < 34 vs. PE > 34 IUGR > 34 vs. N > 34 PE > 34 vs. N > 34 SGA > 34 vs. N > 34
**Maternal age, years ^1^**	30 (28, 35)	30 (28, 34)	34 (30, 36)	35 (35, 37)	32 (27, 35)	34 (32, 34)	0.827 0.468 0.421 0.092 0.598
**Diastolic pressure, mm Hg ^1^**	70 (69, 74)	72 (70, 80)	100 (100, 101)	100 (100, 100)	70 (64, 76)	70 (70, 80)	0.359 1 0.935 **0.003** 0.284
**Systolic pressure, mm Hg ^1^**	112 (110, 120)	116 (113, 120)	160 (150, 164)	163 (155, 173)	117 (108, 119)	120 (110, 120)	0.406 0.416 0.898 **0.004** 0.398
**APGAR 1 min ^1^**	6 (5, 7)	7 (7, 8)	6.5 (6, 7)	8 (7, 8)	8 (8, 8)	8 (8, 8)	**0.003** **0.024** 0.098 0.29 0.677
**APGAR 5 min ^1^**	7 (7, 7)	8 (8, 9)	8 (7, 8)	8 (8, 9)	8.5 (8, 9)	8 (8, 9)	<**0.001** 0.151 0.925 0.762 0.454
**BMI, (kg/m^2^) ^1^**	25 (22, 29)	23 (21, 26)	26 (24, 30)	29 (27, 32)	26 (24, 27)	23 (22, 26)	0.304 0.219 0.873 0.054 0.52
**IFC, CTG ^1^**	0.16 (0.15, 0.61)	0.42 (0.32, 0.67)	0.38 (0.29, 0.55)	0.4 (0.28, 0.57)	0.56 (0.21, 0.95)	0.15 (0.09, 0.18)	0.336 0.832 **0.007** <**0.001** 0.072
**PI of Left Uterine Artery ^1^**	1.03 (0.83, 1.34)	0.91 (0.82, 1.03)	1.06 (0.94, 1.09)	0.95 (0.76, 1.14)	0.85 (0.76, 0.9)	0.91 (0.91, 1)	0.513 0.523 0.936 0.898 0.092
**PI of Right Uterine Artery ^1^**	1.05 (0.85, 1.06)	1.06 (0.98, 1.07)	1.33 (1.27, 1.39)	1.3 (0.92, 1.58)	0.82 (0.78, 0.88)	0.85 (0.8, 0.92)	0.851 0.55 **0.039** 0.095 0.642
**Birth weight, g ^1^**	1144 (921, 1455)	2170 (1960, 2228)	1100 (989, 1421)	2080 (1735, 2617)	2300 (2231, 2375)	2660 (2505, 2700)	<**0.001** <**0.001** <**0.001** 0.259 0.056
**Placenta mass, g ^1^**	225 (216, 251)	216 (216, 265)	287 (270, 287)	287 (281, 324)	348 (322, 365)	352 (352, 392)	0.332 0.48 <**0.001** 0.122 0.323
**PI of umbilical artery ^1^**	1.15 (1, 1.31)	1.05 (0.97, 1.07)	0.96 (0.94, 1.06)	0.96 (0.88, 1)	1.01 (0.92, 1.12)	0.85 (0.8, 0.85)	0.123 0.525 <**0.001** 0.303 **0.048**
**Estimated fetal weight, g ^1^**	1324 (1041, 1611)	1936 (1737, 2114)	1153 (1033, 1595)	1985 (1759, 2462)	2277 (2188, 2349)	2515 (2400, 2617)	<**0.001** <**0.001** <**0.001** 0.097 0.072
**Percentile of estimated fetal weight ^1^**	2.12 (1.52, 2.25)	1 (0.2, 2.3)	15 (3.75, 23.2)	13 (8.7, 25.39)	4.3 (3.38, 6.8)	25 (22.05, 27.75)	0.418 0.966 <**0.001** 0.165 <**0.001**
**Gestational age at the time of delivery, weeks ^1^**	31 (30, 32)	37 (37, 37)	31 (29, 32)	35 (34, 36)	37 (36, 38)	35 (35, 37)	<**0.001** <**0.001** 0.056 0.335 0.063
**PI in MCA ^1^**	1.87 (1.6, 2.13)	1.75 (1.51, 1.87)	1.79 (1.66, 1.79)	1.68 (1.56, 1.86)	1.64 (1.49, 1.8)	1.63 (1.56, 1.63)	0.437 0.965 0.475 0.335 0.684
**Cerebroplacental ratio ^1^**	1.43 (0.99, 1.77)	1.64 (1.54, 1.64)	1.55 (1.51, 1.55)	1.61 (1.55, 1.85)	1.55 (1.41, 1.78)	2.23 (2.03, 2.23)	0.434 0.061 **0.005** <**0.001** **0.023**
**Intraventricular hemorrhage, % ^2^**	2 (20%)	2 (15.4%)	0	0	1 (12.5%)	0	-
**Respiratory distress syndrome, % ^2^**	10 (100%)	1 (7.7%)	12 (100%)	3 (42.9%)	0	1 (14.3%)	-
**Proteinuria, % ^2^**	0	0	10 (83.3%)	4 (57.14%)	0	0	-

^1^ Data are presented as median (Me) and quartiles Q1 and Q3 in the Me format (Q1; Q3) with an indication of the statistical significance (*p*) using the Mann–Whitney test; ^2^ Data are presented as absolute numbers *N* and percentages of the total number of patients in a group P in the format N (P%).

**Table 2 diagnostics-11-00729-t002:** miRNAs and their target genes, differentially expressed in the placental bed samples from women with IUGR < 34 according to the data of miRNA deep sequencing and mass spectrometry.

miRNA	log2(FC) *	*p ***	Gene-Target Name	Protein Name (Protein ID, UniProtKB)	log2(FC) *	*p ***
hsa-miR-1-3p	1.18	0.00012	*ANXA2*	Annexin A2 (P07355)	−2.15	<0.0001
hsa-miR-1-3p	1.18	0.00012	*TAGLN2*	Transgelin-2 (P37802)	−2.088	<0.0001
hsa-miR-1-3p	1.18	0.00012	*YWHAZ*	14-3-3 protein zeta/delta (P63104)	−1.032	0.00193
hsa-miR-1-3p	1.18	0.00012	*FN1*	Fibronectin (P02751)	−6.999	0.00504
hsa-miR-199a-3p	0.72	0.00026	*FN1*	Fibronectin (P02751)	−6.999	0.00504
hsa-miR-199b-3p	0.72	0.00027	*FN1*	Fibronectin (P02751)	−6.999	0.00504
hsa-miR-185-3p	−0.95	0.00060	*FLNA*	Filamin-A (P21333)	0.9255	0.1388
hsa-miR-128-3p	0.80	0.0024	*FN1*	Fibronectin (P02751)	−6.999	0.00504
hsa-miR-140-3p	−0.71	0.0025	*DSTN*	Destrin (P60981)	0.878	0.00531
hsa-miR-30c-5p	0.62	0.0075	*FBN1*	Fibrillin-1 (P35555)	−6.985	0.0689
hsa-miR-30c-5p	0.62	0.0075	*VIM*	Vimentin (P08670)	−1.695	0.0012
hsa-miR-30c-5p	0.62	0.0075	*YWHAZ*	14-3-3 protein zeta/delta (P63104)	−1.032	0.00193
hsa-miR-28-3p	0.56	0.0140	*VIM*	Vimentin (P08670)	−1.695	0.0012
hsa-miR-30d-5p	−0.59	0.02025	*MYH11*	Myosin-11 (P35749)	1.9163	<0.0001
hsa-miR-654-3p	0.57	0.0421	*FGA*	Fibrinogen alpha chain (P02671)	−7.686	0.0002243
hsa-miR-15b-5p	−0.53	0.0495	*MYL6*	Myosin light polypeptide 6 (P60660)	1.5447	0.000172

* log2(FC) = logarithm value to base two of median fold change (FC) in the expression level of miRNA or target gene. ** *p* = statistical significance of the expression level of miRNA or target gene.

**Table 3 diagnostics-11-00729-t003:** Pairwise comparison of groups with IUGR, PE, SGA, and control group N by expression level of miRNA and mRNA obtained in RT-PCR.

miRNA or mRNA	Groups to Be Compared ^1^		*p*-Value
	**IUGR < 34**	**IUGR > 34**	
hsa-miR-1-3p	5.58 (4.71, 6.82)	7.04 (6.28, 8.9)	0.0374
hsa-miR-128-3p	6.9 (5.84, 7.23)	7.87 (6.98, 8.28)	0.0268
YWHAZ	8.76 (8.53, 8.92)	8.07 (7.49, 8.39)	0.0268
ANXA2	1 (0.57, 1.21)	0.66 (0.18, 0.77)	0.0374
FBN1	3.73 (3.28, 3.95)	2.99 (2.58, 3.42)	0.0439
	**PE < 34**	**PE > 34**	
DSTN	6.84 (6.68, 7.13)	7.46 (7.29, 7.49)	0.0026
FGA	11.07 (9.56, 13.47)	8.29 (7.3, 9.52)	0.0221
MYH11	8.52 (8.26, 8.92)	9.39 (9.13, 9.83)	0.0098
MYL6 (tr.var.1)	3.62 (3.46, 3.8)	4.01 (3.92, 4.11)	0.0008
MYL6 (tr.var.2)	3.57 (3.34, 3.72)	4.17 (3.62, 4.38)	0.0358
	**IUGR > 34**	**N > 34**	
hsa-miR-1-3p	7.04 (6.28, 8.9)	5.37 (3.64, 5.96)	0.0221
hsa-miR-140-3p	0.08 (−0.4,0.48)	−0.88 (−1.63, −0.36)	0.0449
hsa-miR-15b-5p	2.12 (1.64, 2.46)	0.77 (0.27, 0.92)	0.0008
hsa-miR-199a-3p	5.21 (4.71, 5.79)	3.89 (2.69, 4.18)	0.0037
hsa-miR-199b-3p	5.26 (4.75, 5.7)	3.58 (2.59, 4.27)	0.0098
hsa-miR-28-3p	5.88 (5.53, 6.75)	4.33 (4.01, 4.8)	0.0001
hsa-miR-30c-5p	5.72 (5.56, 6.42)	4.04 (2.75, 4.6)	0.0026
hsa-miR-654-3p	10.85 (9.52, 11.91)	9.38 (8.49, 9.72)	0.0449
VIM	1.06 (0.94, 1.31)	1.47 (1.39, 1.75)	0.0221
ANXA2	0.66 (0.18, 0.77)	0.85 (0.81, 1.13)	0.0130
FGA	8.59 (5.94, 11.9)	13.48 (12.06, 17.19)	0.0098
MYL 6 tr.v.1	3.84 (3.5, 4.09)	4.06 (3.9, 4.33)	0.0373
MYL 6 tr.v.2	3.62 (3.29, 4.1)	4.03 (3.67, 4.06)	0.0498
YWHAZ	8.07 (7.49, 8.39)	8.7 (8.08, 9.18)	0.0414
FBN1	2.99 (2.58, 3.42)	3.28 (2.82, 3.78)	0.0374
	**SGA > 34**	**N > 34**	
FGA	8.67 (7.53, 12.52)	13.48 (12.06, 17.19)	0.0229
MYL 6 tr.v.1	3.67 (3.54, 4.11)	4.06 (3.9, 4.33)	0.0274
hsa-miR-654-3p	9.74 (9.06, 12.25)	9.38 (8.49, 9.72)	0.0269
hsa-miR-15b-5p	1.09 (0.95, 1.39)	0.77 (0.27, 0.92)	0.0453
hsa-miR-185-3p	7.87 (7.23, 8.16)	6.73 (6.36, 7.49)	0.0302
hsa-miR-140-3p	0.06 (−0.2, 0.43)	−0.88 (−1.63, −0.36)	0.0189
	**PE > 34**	**N > 34**	
hsa-miR-30c-5p	5.82 (4.95, 6.36)	4.04 (2.75, 4.6)	0.0379
hsa-miR-654-3p	9.89 (9.27, 10.99)	9.38 (8.49, 9.72)	0.0494
hsa-miR-185-3p	7.88 (7.45, 8.2)	6.73 (6.36, 7.49)	0.0433
FGA	8.29 (7.3, 9.52)	13.48 (12.06, 17.19)	0.0023
FLNA	7.85 (7.27, 7.87)	8.13 (7.63, 8.29)	0.0498
YWHAZ	8.02 (7.76, 8.43)	8.7 (8.08, 9.18)	0.0493
MYH11	9.39 (9.13, 9.83)	8.32 (7.77, 8.96)	0.0281

^1^ Data are presented as a median (Me) of the ΔCt values and quartiles Q1 and Q3 in the Me (Q1; Q3) format with an indication of the statistical significance (*p*) using the Mann–Whitney test.

**Table 4 diagnostics-11-00729-t004:** Pairwise comparison of the IUGR, PE, SGA, and N groups by the expression level ratio of miRNA and its target gene, obtained by RT-PCR in each of the studied placental bed samples.

miRNA/mRNA Pair	Groups to Be Compared ^1^		*p*-Value
	**IUGR > 34**	**N > 34**	
miR-28-3p/VIM	−4.74 (−5.79, −4.22)	−2.49 (−3.48, −2.43)	0.001
miR-654-3p/FGA	−0.81 (−3.93, 0.83)	4.43 (2.46, 9.85)	0.006
miR-15b-5p/MYL6 tr.v.2	2.13 (1.02, 2.5)	3.32 (2.7, 3.72)	0.006
miR-15b-5p/MYL6 tr.v.1	1.57 (1.53, 2.46)	3.29 (2.87, 3.9)	0.0007
miR-1-3p/ANXA2	−5.78 (−8.25, −5.23)	−4.13 (−5.04, −2.61)	0.037
miR-1-3p/YWHAZ	1.09 (−0.55, 2.43)	4.28 (2.39, 5.73)	0.03
miR-30c-5p/FBN1	−2.73 (−3.7, −2.55)	0.24 (−1.53, 0.79)	0.02
miR-30c-5p/YWHAZ	2 (0.89, 2.63)	5.81 (3.84, 6.43)	0.006
miR-30c-5p/VIM	−4.61 (−5.18, −4.32)	−2.37 (−3.2, −1.18)	0.006
	**SGA > 34**	**N > 34**	
miR-654-3p/FGA	−1.46 (−2.5, 1.18)	4.43 (2.46, 9.85)	0.006
	**PE > 34**	**N > 34**	
miR-654-3p/FGA	−1.84 (−2.88, 0.1)	4.43 (2.46, 9.85)	0.001
miR-185-3p/FLNA	−0.49 (−0.66, 0.13)	1.4 (0.53, 1.93)	0.0026
miR-30c-5p/YWHAZ	2.82 (1.55, 3.39)	5.81 (3.84, 6.43)	0.0026
	**IUGR > 34**	**SGA > 34**	
miR-30c-5p/VIM	−4.61 (−5.18, −4.32)	−3.16 (−3.78, −2.87)	0.016
miR-15b-5p/MYL6 tr.v.1	1.57 (1.53, 2.46)	2.58 (2.1, 3.06)	0.01
miR-30c-5p/FBN1	−2.73 (−3.7, −2.55)	−1.62 (−1.98, −0.98)	0.02
	**PE > 34**	**IUGR > 34**	
miR-15b-5p/MYL6 tr.v.2	2.8 (2.74, 3.29)	2.13 (1.02, 2.5)	0.001
miR-15b-5p/MYL6 tr.v.1	3.22 (2.31, 3.29)	1.57 (1.53, 2.46)	0.0046

^1^ Data on a logarithm to base two of the expression level ratio of miRNA and its gene target mRNA, presented as a median of (ΔCt(mRNA) − ΔCt(miRNA)) values, and quartiles Q1 and Q3 in the Me (Q1; Q3) format with an indication of the statistical significance (*p*) using the Mann–Whitney test.

**Table 5 diagnostics-11-00729-t005:** Metascape Enrichment Analysis.

ID	Category	Description	Hits ^1^	Lg(P)	Lg(Q)
1	GO Biological Processes	supramolecular fiber organization	**ANXA2**|FLNA|MYH11|**VIM**|DSTN	−5.35	−2.35
2	Reactome Gene Sets	Signaling by Interleukins	**ANXA2**|FN1|**VIM**|**YWHAZ**	−4.66	−1.86
3	Reactome Gene Sets	RHO GTPases activate PAKs	FLNA|MYH11|**MYL6**	−7.04	−3.42
4	Reactome Gene Sets	Platelet activation, signaling and aggregation	**FGA**|FLNA|FN1|**YWHAZ**|TAGLN2	−7.51	−3.42
5	GO Biological Processes	blood vessel development	**ANXA2**|FN1|**YWHAZ**	−2.50	0.00
6	GO Biological Processes	blood vessel morphogenesis	**ANXA2**|FN1|**YWHAZ**	−2.64	−0.07
7	GO Biological Processes	protein localization to membrane	**ANXA2**|FLNA|**YWHAZ**	−2.74	−0.15
8	GO Biological Processes	angiogenesis	**ANXA2**|FN1|**YWHAZ**	−2.82	−0.22
9	GO Biological Processes	positive regulation of organelle organization	**ANXA2**|FLNA|**YWHAZ**|DSTN	−4.10	−1.36
10	GO Biological Processes	actin cytoskeleton organization	FLNA|MYH11|DSTN	−2.64	−0.07
11	GO Biological Processes	actin filament-based movement	FLNA|**MYL6**|**VIM**	−4.64	−1.85
12	GO Biological Processes	muscle system process	FLNA|MYH11|**MYL6**|**VIM**	−4.66	−1.86
13	Reactome Gene Sets	Signaling by Rho GTPases	FLNA|MYH11|**MYL6**|**YWHAZ**	−4.68	−1.86
14	GO Biological Processes	actin-mediated cell contraction	FLNA|**MYL6**|**VIM**	−4.93	−2.06
15	GO Biological Processes	muscle contraction	FLNA|MYH11|**MYL6**|**VIM**	−5.09	−2.17
16	Reactome Gene Sets	RHO GTPases activate PKNs	MYH11|**MYL6**|**YWHAZ**	−5.21	−2.26
17	Reactome Gene Sets	RHO GTPase Effectors	FLNA|MYH11|**MYL6**|**YWHAZ**	−5.25	−2.28
18	Reactome Gene Sets	Muscle contraction	**ANXA2**|MYH11|**MYL6**|**VIM**	−6.04	−2.89
19	Reactome Gene Sets	Smooth Muscle Contraction	**ANXA2**|MYH11|**MYL6**	−6.35	−3.09
20	GO Biological Processes	platelet activation	**FGA**|FLNA|FN1|**YWHAZ**	−6.49	−3.19
21	GO Biological Processes	extracellular matrix organization	**ANXA2**|**FBN1**|**FGA**|FN1|MYH11	−6.62	−3.27
22	Reactome Gene Sets	Response to elevated platelet cytosolic Ca2+	**FGA**|FLNA|FN1|TAGLN2	−6.79	−3.40
23	Reactome Gene Sets	Platelet degranulation	**FGA**|FLNA|FN1|TAGLN2	−6.86	−3.42
24	GO Biological Processes	platelet degranulation	**FGA**|FLNA|FN1|TAGLN2	−6.86	−3.42
25	GO Biological Processes	coagulation	**ANXA2**|**FGA**|FLNA|FN1|**YWHAZ**	−6.89	−3.42
26	GO Biological Processes	hemostasis	**ANXA2**|**FGA**|FLNA|FN1|**YWHAZ**	−6.90	−3.42
27	GO Biological Processes	blood coagulation	**ANXA2**|**FGA**|FLNA|FN1|**YWHAZ**	−6.93	−3.42
28	Reactome Gene Sets	Hemostasis	**ANXA2**|**FGA**|FLNA|FN1|**YWHAZ**|TAGLN2	−7.31	−3.42

^1^ Proteins with differential expression in the IUGR group relative to the control group by quantitative PCR are highlighted in bold.

**Table 6 diagnostics-11-00729-t006:** miRNA and mRNA sequence parameters.

miRNA or mRNA	miRNA or mRNA Accession Number (miRbase, Unigene)	Nucleotide Sequence of PCR Primer, 5’-3’	PCR Primers Annealing Temperature, °C
hsa-miR-1-3p	MIMAT0000416	TGGAATGTAAAGAAGTATGTAT	52.7
hsa-miR-199a-3p	MIMAT0000232	ACAGTAGTCTGCACATTGGTTA	57.6
hsa-miR-199b-3p	MIMAT0004563	ACAGTAGTCTGCACATTGGTTA	57.6
hsa-miR-185-3p	MIMAT0004611	AGGGGCTGGCTTTCCTCTGGTC	48.9
hsa-miR-140-3p	MIMAT0004597	TACCACAGGGTAGAACCACGG	48.9
hsa-miR-30c-5p	MIMAT0000244	TGTAAACATCCTACACTCTCAGC	57.6
hsa-miR-654-3p	MIMAT0004814	TATGTCTGCTGACCATCACCTT	52.7
hsa-miR-15b-5p	MIMAT0000417	TAGCAGCACATCATGGTTTACA	57.6
hsa-miR-128-3p	MIMAT0000424	TCACAGTGAACCGGTCTCTTT	59
hsa-miR-28-3p	MIMAT0004502	CACTAGATTGTGAGCTCCTGGA	54
hsa-miR-30d-5p	MIMAT0000245	TGTAAACATCCCCGACTGGAAG	54
SNORD68	NR_002450	Hs_SNORD68_11 miScript Primer Assay, Qiagen	55
ANXA2 F	NM_004039.3	CAGCATTTGGGGACGCTCTCA	48.9
ANXA2 R		AATGGTGACCTCATCCACACC	
TAGLN2 F	NM_003564.3	CCCTCACTGTGCTGCTCTTT	52.7
TAGLN2 R		GCCATCCTTGAGCCAGTTCT	
YWHAZ F	NM_003406.4	ACGACGTCCCTCAAACCTTG	52.7
YWHAZ R		TGACCTACGGGCTCCTACAA	
FN1 F	XM_017003692.1	TGGTGCCATGACAATGGTGT	48.9
FN1 R		CGGGAATCTTCTCTGTCAGCC	
FLNA F	NM_001456.4	AGTGTCAATCGGAGGTCACG	61.6
FLNA R		CTGGTCACATCCAGCCCATT	
DSTN F	NM_001011546.2	AGTGTTGGGCCAGGCTTTAG	57.6
DSTN R		TGCATCCTTGGAGCTTGCAT	
FBN1 F	NM_000138.5	ACATCTCCGCGTGTATCGAC	57.6
FBN1 R		CACAGGTCCCACTTAGGCAG	
VIM F	NM_003380.5	GGACCAGCTAACCAACGACA	46.2
VIM R		AAGGTCAAGACGTGCCAGAG	
MYH11 F	NM_002474.3	AGCGGTACTTCTCAGGGCTA	52.7
MYH11 R		AGATGGGCCTTGCGTGATAC	
FGA F	NM_000508.5	GCAAAGATTCAGACTGGCCC	61.6
FGA R		CTACTGCATGACCCTCGACA	
MYL6 tr.v.1 F	NM_021019.5	ACTTCACCGAAGACCAGACC	57.6
MYL6 tr.v.1 R		TATGCCTCACAAACGCTTCATAG	
MYL6 tr.v.2 F	NM_079423.4	ACTTCACCGAAGACCAGACC	52.7
MYL6 tr.v.2 R		GCGGACGAGCTCTTCATAGT	
GAPDH F	NM_002046	ACCACAGTCCATGCCATCAC	60
GAPDH R		TCCACCACCCTGTTGCTGTA	
ACTB F	NM_001101	GGACTTCGAGCAAGAGATGG	60
ACTB R		AGCACTGTGTTGGCGTACAG	

## Data Availability

The data presented in this study are available in Appendix A and Appendix A.

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
