# Peer review of "miRNAs and Their Gene Targets—A Clue to Differentiate Pregnancies with Small for Gestational Age Newborns, Intrauterine Growth Restriction, and Preeclampsia"

_diagnostics, 2021, doi:10.3390/diagnostics11040729_

Round 1

Reviewer 1 Report

The current study entitled "miRNAs and their Gene Targets—A Clue to Differentiate Pregnancies with Small for Gestational Age Newborns, Intrauterine Growth Eestriction, and Preeclampsia" adds significant knowledge regarding differential miRNA expression in the placenta and placental bed tissues isolated form PE, IUGR and SGA pregnancies. The authors found that the main changes in transcriptome and proteome profiles in IUGR occurred in the placental bed, in contrast to PE, in which the majority of molecular changes occurred in the placenta. 

The study is mostly well done and presented, however, a few points need to be addressed-

1- The title mentions "Intrauterine Growth Estriction" which should be "Intrauterine Growth Restriction"

2- The study is primarily focused on miRNA but the introduction provides very little detail. There must be some introductory information as well as a description of previous key studies. Similar to small noncoding RNAs (sncRNAs) piRNA also needs some additional details.

3- The exclusion criteria were mentioned as multifetal pregnancies, severe somatic pathology in pregnant women, fetal aneuploidy, and vaginal delivery. Is there any information available regarding prescription or nonprescription drug usage during pregnancy that may affect the miRNA expression profile.

Author Response

Dear Reviewer #1. I received your letter with comments on our manuscript “miRNAs and their Gene Targets—A Clue to Differentiate Pregnancies with Small for Gestational Age Newborns, Intrauterine Growth Restriction, and Preeclampsia" (diagnostics-1093744). I appreciate the time you spent reviewing the manuscript and I am very grateful for your detailed analysis of the data presented. Thank you very much for your opinion. I have corrected the article in light of your comments and recommendations.

  • The title mentions "Intrauterine Growth Estriction" which should be "Intrauterine Growth Restriction"

The spelling error has been corrected.

  • The study is primarily focused on miRNA but the introduction provides very little detail. There must be some introductory information as well as a description of previous key studies. Similar to small non-coding RNAs (sncRNAs) piRNA also needs some additional details.

The introduction has been supplemented with the information concerning the previously performed systematic analysis of the possible differences in the pathogenesis of preeclampsia and IUGR [review article of Aplin JD, 2020], and on the function and participation of small non-coding RNAs in the occurrence and development of these complications of pregnancy [review articles of Carvalho Barbosa, 2020; Hu X, 2019].

  • The exclusion criteria were mentioned as multifetal pregnancies, severe somatic pathology in pregnant women, fetal aneuploidy, and vaginal delivery. Is there any information available regarding prescription or nonprescription drug usage during pregnancy that may affect the miRNA expression profile?

The Materials and Methods section (subsection 4.1) has been supplemented with the information concerning prescription and nonprescription drug usage during pregnancy in each studied group. The likelihood of the influence of low molecular weight heparins and antihypertensive drugs on the miRNA expression profile in the placenta and the placental bed is minimal since the effect of these drugs prescribed in the third trimester of pregnancy is symptomatic rather than pathogenetic.

Reviewer 2 Report

The authors report a series of studies examining miRNA expression in pregnancies affected by PE and FGR. This is a well-designed and reported study and I have no major concerns that should delay publication. I have made some minor comments below.

Introduction:

Clearly written hypothesis and aims of study clearly described.

Methods:

The GA for group 2 is not described in the text.

Group 7 seems very heterogeneous w.r.t aetiologies.

The description of lab methodology is comprehensive.

The description of statistical analysis is also comprehensive.

Results:

This is long but clearly sub-divided and easy to follow.

Discussion:

This is a very comprehensive review of the results and relevant literature.

There is no discussion of the strengths and limitations of the study.

There is no discussion of how these findings might be applied to the clinical situation.

Author Response

Dear Reviewer #2. I received your letter with comments on our manuscript “miRNAs and their Gene Targets—A Clue to Differentiate Pregnancies with Small for Gestational Age Newborns, Intrauterine Growth Restriction, and Preeclampsia" (diagnostics-1093744). I appreciate the time you spent reviewing the manuscript and I am very grateful for your detailed analysis of the data presented. Thank you very much for your opinion. I have corrected the article in light of your comments and recommendations.

  • The GA for group 2 is not described in the text.

This information has already been indicated in section 4.1.

  • Group 7 seems very heterogeneous w.r.t aetiologies.

We agree with you that there are many reasons for premature birth earlier than 34 GW in the absence of PE and IUGR, and it is the case often due to pelvic inflammatory disease. That is why samples from women with premature birth are not optimal but the only possible controls for age-matched samples from pregnant women with IUGR or PE. Therefore, we considered it more informative and correct to compare samples of placental bed samples from healthy women, delivered after 34 GW (N > 34), with age-matched samples from women with IUGR > 34, PE > 34, or SGA > 34, in order to understand the molecular mechanisms of the pathogenesis of pregnancy complications. Moreover, since all women from the SGA group were delivered after 34 GW, the groups with IUGR > 34, PE > 34, and control group N > 34 were selected, but not other groups (IUGR < 34, PE < 34, N < 34), for comparison with the SGA group in the same GW range.

  • There is no discussion of the strengths and limitations of the study.

These aspects are presented at the end of the Discussion section.